# Unqualified Advice and Product Promotions: Analysis of Health and Nutrition Content on Social Media Consumed by Young Adults

**DOI:** 10.3390/nu18010044

**Published:** 2025-12-22

**Authors:** Sophie Evans, Kelly Lambert, Adrian Dinale, Myah Quinn, Denelle Cosier

**Affiliations:** School of Medical, Indigenous and Health Sciences, University of Wollongong, Wollongong, NSW 2500, Australia; sme412@uowmail.edu.au (S.E.); maq694@uowmail.edu.au (M.Q.); dcosier@uow.edu.au (D.C.)

**Keywords:** social media, Instagram, eating behaviours, thematic analysis, TikTok

## Abstract

Background/Objectives: This study investigated the relationship between time spent on social media and eating behaviours among young Australian adults. It also examined the types of content discussed and linguistic styles used by health and nutrition content creators on Instagram. Methods: Young adults (aged 18–30 years) who reported viewing social media for nutrition or health content were recruited to complete a self-administered, cross-sectional survey. Data on demographics, time spent on Instagram and TikTok, health content creators viewed, and responses to the Scale of Effects of Social Media on Eating Behaviours (SESMEB) were collected. Associations between time spent on Instagram and TikTok and SESMEB scores were analysed. Inductive content and thematic analysis were conducted on health-related posts from Instagram accounts viewed by study participants. Results: From the 57 participants who completed the demographic survey, 42 participants completed the full study including the SESMEB survey. There was no significant association between SESMEB score and time spent on Instagram (*p* = 0.38) or TikTok (*p* = 0.40). A total of 1420 Instagram posts from 71 distinct content creators were analysed. Health and fitness product endorsements or advertisements (56.3%), predominantly posted by laypersons (55.3%), were the most common type of post in the sample. The most common communication style was ‘expert advice’ (47.9%), with ‘informal language’ (85.9%) as the dominant linguistic style. Results from thematic analysis suggest health and nutrition information on social media is often presented to consumers in emotionally charged, stylised, or contradictory ways and requires users to sift through conflicting messages, aesthetics, and ideologies to construct their own understanding of health. Conclusions: This study suggests that young adults are primarily exposed to health and fitness product promotions from unqualified content creators on social media. Dietitians and nutrition professionals may need to consider adopting specific linguistic and communication styles to enhance the dissemination and engagement of credible nutrition information online. These findings have implications for improving digital health literacy and strengthening the impact of evidence-based nutrition messaging in digital environments.

## 1. Introduction

Social media use is regularly used by 63.9% of the global population. Among young Australian adults, usage is even higher, with 85% [1] engaging in social media for an average of two hours a day [2]. As access to social media increases, social media platforms have become a fundamental medium for communication in society [3]. The virtual landscape of social media has evolved into platforms used for sharing information and education [4], particularly regarding food and nutrition [4]. Previous research has shown that users not only actively search for but are also passively exposed to popular food and nutrition content [5].

Previous research has found that social media use may influence individuals’ dietary behaviours, particularly amongst young adults [6,7]. A recent systematic review reported that longer duration on various social media platforms correlated with increased reports of dieting and unhealthy food choices [8]. However, the effect of social media on eating behaviours is also related to the type of content viewed, with some content types influencing positive dietary habits and health outcomes and others influencing adverse dietary habits and health outcomes [9]. Recent research has found that viewing fitness inspiration images (‘fitspiration’) or other body image-related content idealising certain appearances is associated with heightened body comparison and influences food choices in young adults [7,8]. Conversely, other studies have found that using social media for recipes and nutrition advice increased nutrition knowledge and most consistently predicted positive dietary behaviour changes among young Australian adults [10,11].

Engagement with health and nutrition content is most prevalent on Instagram [12] and TikTok [13], the third and fifth most used social media platforms in Australia [14]. Due to the popularity of health and nutrition information delivered as image- and video-based content, such apps have been recognised as a potential tool for health promotion [15]. However, previous health campaigns that attempt to leverage social media to target young adults have resulted in poor engagement [16]. Instead, the rise of the health and wellness media culture has created an online environment where food and nutrition information distribution is dominated by industry marketing [12] and amplified by health and lifestyle ‘influencers’ [17]. In addition, recent evidence suggests that the nutrition and health content promoted is frequently inaccurate and not provided by health professionals [18]. An exploratory study found that 44.8% of nutrition information posted by popular Instagram accounts contained inaccuracies, with brand accounts and fitness influencers having a higher likelihood of publishing misleading content [19]. Furthermore, the continual onslaught of new health trends promoted by influencers on social media makes it additionally difficult for laypeople to differentiate between evidence-based messages and misinformation or qualified health experts from unqualified individuals [19,20]. A recent study found that the credibility of social media health information is influenced by the language used, perceived expertise, and popular social media metrics, such as the number of views a video receives [21].

Given that previous studies have shown negative impacts of social media on eating behaviours among young adults [8], and frequent exposure to inaccurate health information [10], it is essential to explore the impact of social media use on eating behaviours among Australian young adults. To our knowledge, no previous Australian studies have examined time spent on social media amongst young adults and whether this is related to their eating behaviours. Additionally, no previous Australian studies have conducted a thorough content and thematic analysis of health and nutrition posted content on both platforms, Instagram and TikTok. No previous research has examined the specific accounts of content creators reported by participants that are identified as influencing their eating behaviours. Therefore, the aim of this study was to examine whether there is a relationship between time spent on social media and eating behaviours amongst Australian young adults. Additionally, we aimed to explicitly examine the content of health and nutrition social media posts from influencers followed by study participants. We hypothesised that there was a positive relationship between time spent on social media and eating behaviours amongst study participants.

## 2. Materials and Methods

A web-based, cross-sectional, sequential exploratory mixed-methods study was conducted between January and June 2025. The study is designed with two parts: (i) collection of survey data from questionnaires; (ii) content and thematic analysis of selected posts from Instagram and TikTok. Inclusion criteria were adults aged 18–30 years old who (1) live in Australia, (2) use Instagram and/or TikTok, and (3) view nutrition and health content on Instagram. Based on prior exploratory studies [10,22], the minimum sample size required was 50–100 participants [10,22]. Participants were recruited using convenience sampling. Recruitment posters were placed around the University of Wollongong (UOW), on student Facebook pages, and circulated through the research team’s professional networks.

A self-administered web-based survey was developed for the study. The survey collected demographic information, social media use, and the Scale of Effects of Social Media on Eating Behaviour (SESMEB) [22]. Demographic information included sex, age, education level, area of post-school qualification, current occupation, whether the individual followed a specific diet, and postcode of residence to determine the Socioeconomic Index For Areas (SEIFA) advantage or disadvantage rating [23]. Information on the average time spent per day on Instagram and TikTok was collected. Participants who accessed social media platforms on their smartphones were encouraged to use the screen time reports available in the settings application for an accurate report of time spent on each platform. Participants were also asked to report up to three health influencer accounts that they follow on Instagram. The use of Instagram and TikTok was chosen as the focus in this study due to research indicating a high level of engagement with health and nutrition content on these two platforms amongst adults [12,15].

The final survey section included the Scale of Effects of Social Media on Eating Behaviour (SESMEB) questionnaire, a valid and reliable tool for assessing the level of influence that social media may have on individual eating behaviours [22]. The SESMEB questionnaire consists of a five-point Likert scale with responses ranging from ‘always’ (five points) to ‘often’ (four points), ‘sometimes’ (three points), ‘rarely’ (two points), and ‘never’ (one point). Total scores were calculated by adding the score from each of the 18 items, ranging from 18 to 90, with a higher total score indicating a higher level of influence by social media on eating behaviours [22].

Statistical analyses were conducted in Jamovi (version 2.5) [24]. Normality of data was tested using the Shapiro–Wilk test. Descriptive statistics were reported as mean ± SD and frequencies. Associations between time spent on each social media platform and total SESMEB scores were analysed using one-way ANOVA. Associations between demographic variables and the total SESMEB scores were assessed using one-way ANOVA for categorical variables and Pearson’s correlation for continuous variables.

Content and thematic analysis were conducted on the 20 most recent Instagram posts from each health Instagram account nominated by study participants (Figure 1). Researchers accessed Instagram using personal accounts to capture static screenshots of images and videos and recorded the accompanying captions, hashtags, publication dates, audio transcripts, artist names, and any music associated with the posts. Content creators who did not share health- or nutrition-related information were excluded from the analysis. Four researchers independently screened the posts, with one researcher cross-screening for consistency. Discrepancies in categorisation were discussed among three team members and excluded when appropriate. Data for each content creator was compiled into a Word document and stored on Microsoft Teams.

Content analysis was used to examine and categorise the types of content shared by the nominated Instagram creators. This stage of content analysis included identifying content topics, creator types, communication styles [25], and linguistic features [26]. Frequencies of codes were reported. Codes were not mutually exclusive, that is creators could be assigned to multiple categories where relevant. All coding was conducted and stored in Microsoft Excel.

A three-phase inductive analytical approach was undertaken by one researcher to categorise content topics [27]. First, the researcher reviewed the data repeatedly to become immersed in the content and reflect on its subject matter. Second, for initial coding, descriptive codes were generated for each creator based on the theme information present in their 20 posts. Third, similar codes were grouped into higher-order categories to identify overarching content types for each creator. Content creators were categorised based on their professional background into the following groups: layperson, nutritionist, dietitian, fitness professional, certified medical professional, government organisation, and medical doctor. These categories were developed after reviewing the creators nominated by participants and identifying whether they had relevant evidence-based health, nutrition, or dietetic educational qualifications. Content creators were categorised as laypersons when they had not listed any educational qualifications, for example, a university degree, in their Instagram biography.

Each creator’s predominant communication style was classified using a previously published taxonomy of influencer communication styles [25]. This published taxonomy was developed by analysing 11,000 videos from 151 top YouTube influencers using native language processing to identify seven key communication styles used by influencers: intimate experience, struggle and overcoming, motivation and guidance, middle-of-the-road, expert advice, coaching and mentoring, and storytelling [25]. These communications styles utilised by each content creator were identified by one researcher based on tone, language, and audience engagement strategies. Creators may have used multiple communication styles across their analysed posts.

Instagram posts were also coded for the presence or absence of specific linguistic features, based on prior research identifying the language, themes, and rhetorical strategies used by social media influencers [26]. Features assessed included informal language, profanity, jargon, slang, rhetorical questions, non-conformity, individualism, masculinity, entrepreneurship, storytelling, humour, shock value, and authenticity claims. Each Instagram post was coded for the presence/absence of these linguistic features by one researcher; therefore, multiple features could have been utilised in each post.

An inductive thematic analysis approach [28] was used to explore underlying themes and meanings in the Instagram posts. The process included immersion with repeated reading of content and reflection on its meaning. In vivo coding was then undertaken with generation of descriptive codes based on key messages from each creator’s 20 posts, considering both manifest and latent meanings. In vivo coding was performed by two researchers. A codebook was developed at this stage to support deductive categorisation. Similar codes were grouped into higher-order categories to identify overarching concepts by both researchers. Unique codes were retained as outliers. The two researchers reviewed the categories and collaboratively developed overarching themes through consensus to support rigour and credibility [28].

Consent for this study was obtained from the University of Wollongong Human Research Ethics Committee (2024/348). Participants were required to provide informed consent before proceeding with completing the survey questions. Participants were also able to close the web browser at any point in the survey and choose not to proceed if they decided they did not want to participate thereafter. Data from content creators was sourced from public domains and, therefore, did not require consent to be used. However, the still images from the collected posts did not include the account names to maintain the privacy and anonymity of the content creators. All data collected from survey participants was anonymous, and all data collected from social media accounts were de-identified.

## 3. Results

Of the 128 individuals who commenced the survey, 57 (44%) completed demographic information and 42 (33%) provided complete responses to the survey (Figure 2).

### 3.1. Participant Characteristics

Most (67.9%) participants identified as female, and the mean age of the sample was 21.1 ± 2.94 years. Almost half (49.1%) of participants were residents of high socioeconomic level suburbs indicated by SEIFA deciles 8–10. Most individuals had completed Year 12 or an equivalent qualification (*n* = 32, 56.1%), followed by a bachelor’s degree (*n* = 19, 33.3%). Of the 50 participants who selected that they have or are completing post-school qualifications, the areas included health (*n* = 26, 49.1%), society and culture (*n* = 6, 11.3%), and education (*n* = 4, 7.5%). Among those employed, the most common occupations were hospitality (*n* = 17, 36.2%) and retail and sales (*n* = 11, 23.1%).

A total of 23 participants (54.8%) reported following a special diet, including a reduced-calorie/calorie-deficient diet (*n* = 4, 9.5%) and a Mediterranean diet (*n* = 3, 7.1%). Refer to Table 1 below for the complete demographic characteristics.

### 3.2. Time Spent on Social Media

The majority (41.9%) of participants spent an average of 1–2 h per day using Instagram, followed by 39.5% who spent ≤60 min, and 18.6% who spent more than 2 h. No significant associations were found between the amount of time spent using Instagram and demographic characteristics such as age and sex.

The SESMEB tool was used to determine whether time spent on social media was associated with eating behaviours. The total mean SESMEB score was 41.6 ± 9.87 (range, 23–63). There was no significant difference between the amount of time spent using Instagram and the SESMEB score [f = 0.768, *p* = 0.38]. Similarly, there was no significant difference between time spent using TikTok and the SESMEB score [f = 0.996, *p* = 0.40]. Across the 18 items assessing the influence of social media on eating behaviours, participants reported the highest agreement with the statements “The foods/meals that I see on social media stimulate my desire to eat” at a mean score 3.36 ± 0.93, and “When I see food on social media it influences my desire to eat” at a mean score of 3.22 ± 0.97. Another SESMEB item with high agreement included the statement “I follow nutrition news/blogs/pages on social media” with a mean score of 2.98 ± 1.18. In contrast, the SESMEB items with the lowest agreement included the statement “I think foods/meals with more likes are healthier” with a mean score of 1.74 ± 0.93, and the statement “Even though I am full, I eat a food/meal I see on social media” with a mean score of 1.89 ± 0.75. See Figure 3 for mean responses to each item related to the effects of social media on eating habits.

### 3.3. Content Analysis of Social Media Posts

The survey respondents nominated 86 individual content creators, and 71 creators were included in the analysis. This resulted in the inclusion of 1420 individual posts, extracted between 1 April and 18 June 2025. A total of 33 codes were derived from 254 initial codes of the type of content posted by creators. The five most frequently occurring content types amongst the 71 creators were health/fitness product endorsement/advertisements (56.3%), nutrition education and dietary advice (46.5%), healthy recipes and meal inspiration (42.6%), promotion of personal health/fitness products (38%), and fitness motivation (29.6%). Refer to Table 2 below for the complete list of content types from the included social media creators.

The extracted content was then categorised according to the creator’s qualification. Laypeople without evident qualifications were the most common content creators (55.3%), followed by nutritionists (15.5%), fitness professionals (12.7%), and dietitians (11.3%) (Table 3). The communication styles of these creators were then analysed and categorised into seven communication styles [25]. Expert advice (47.9%) was the most frequently used communication style among almost half of content creators. Following in frequency was motivation and guidance (45.1%) and intimate experience (36.6%). The linguistic styles of posts were examined and categorised into the categories shown below in Table 3. The use of informal language was highly prolific amongst creators (85.9% of posts). Similarly, rhetorical questions were used by over three-quarters of posts (84.5%), and jargon/slang was popularly utilised by creators (81.7% of posts). The majority of creators included an element of entrepreneurship in their posts (56.3%). See to Table 3 for further details on the types of creator, communication styles, and linguistic features.

### 3.4. Thematic Analysis of Social Media Posts

Four themes and one overarching concept were evident in the analysis of health and nutrition content. The overarching concept regarding health and nutrition content was that social media posts are often presented to consumers in emotionally charged, stylised, or contradictory ways. This required consumers to sift through conflicting messages, aesthetics, and ideologies to form their own understanding of health and nutrition as part of their health journey.

Four interrelated themes were apparent. Emotional and psychological content was used to evoke feelings, memories, and psychological responses, shaping engagement. For example, creators talked about showing up for yourself, removing guilt from food, and becoming connected through exercise. See Table 4 for example codes from analysed Instagram posts.

Second, influencers presented themselves as knowledgeable and authoritative, with content positioned as educational, insider, or revelatory, often including conflicting, contradictory, or pseudo-scientific claims. Example quotes from codes include “Obviously I’m not female, but I’ve done a bit of research” and “If you consistently under-consume calories, the body can downregulate processes like digestion to conserve energy”.

The third theme identified was social and cultural framing of content. Influencers used language, aesthetic features, and social cues to align with or challenge cultural norms about food, nutrition, and health. Influencers spoke often about moving away from strict eating and labelling foods as good or bad in quotes including “Give yourself permission to include all foods, because food is more than just fuel”. Influencers additionally engaged in nutrition myth busting utilising ‘insider knowledge’ illustrated by the following quote: “Carbs don’t make you fat. Whoever told you that is lying.”

Finally, social media content was designed to address the practicalities and logistics of managing one’s health and nutrition in everyday life. Practicality and lifestyle integration were communicated as central to success with factors such as convenience, the need for a support team, achieving balance, and a focus on nutrients as subthemes. This was apparent in the following quote: “Healthy eating is not black and white: it’s about balance and flexibility. Some days you won’t have the energy to cook a fresh, well-balanced meal and that’s ok!”

## 4. Discussion

This mixed-methods study aimed to examine whether Instagram use, measured explicitly as average daily time spent on Instagram, was associated with eating behaviours amongst Australian young adults and to investigate the type and nature of the health content viewed by study participants. The qualitative and quantitative nature of this study yielded three primary findings. Firstly, there was no association between the amount of time spent using Instagram or TikTok and SESMEB scores. This null relationship suggests that, in this sample, the duration of time spent on social media platforms did not influence the eating behaviours of participants. Secondly, content analysis identified that, of the 71 content creators identified by study participants, the most common type of content posted was health and fitness product endorsement and advertising, predominantly presented by laypeople. In addition, creators utilised various persuasive devices and communication styles, drawing on expert advice as well as motivation and guidance. The most employed linguistic features were informal language, rhetorical questions, and jargon/slang. Finally, thematic analysis found that the health and nutrition content on social media is emotionally charged, stylised, or contradictory, requiring consumers to navigate the online health space to form their own understanding on their health journey. Collectively, the findings of this study reveal in detail the type of content and style of messaging that young adults viewing health and nutrition information on social media are most frequently exposed to.

Contrary to our hypothesis, we found no association between daily time spent using Instagram and eating behaviours in the study sample. This contrasts with previous research, as a recent systematic review found that greater exposure to content on social media was associated with higher body dissatisfaction, dieting/restricting food, and unhealthy food choices [9] Furthermore, a previous study amongst a large cohort of undergraduate university students in Turkey found that a longer duration spent using social media was associated with higher SESMEB scores [29]. The lack of association between time spent on social media and eating behaviours in our study may indicate that the impact of social media is complex and nuanced and related to the type of content and persuasive techniques individuals are exposed to rather than the magnitude of exposure alone. This is aligned with recent research, finding that different types of linguistic techniques and content styles are associated with different degrees of engagement by viewers [25,30]. Additionally, a systematic review has previously found that higher educational attainment is associated with greater digital literacy and may reduce susceptibility to misleading health information on social media [31]. Given that a significant proportion of participants in the current study indicated that they have or are currently completing post-school qualifications, this may suggest our sample had a reasonably high digital literacy and therefore may be less impacted by what they see on social media. Future studies including a more diverse study population with differences in educational attainment and socioeconomic levels will be beneficial to build on the findings of this study.

The findings of this study revealed that young Australian adults are primarily exposed to product endorsements and advertisements from health and fitness creators. This high frequency of product endorsement raises concerns, given that previous research has shown that Instagram posts about supplements were likely to be of lower accuracy and quality compared to posts about general healthy eating or fitness [18]. Additionally, our study identified that laypeople without evident formal qualifications were the most frequent type of influencer viewed. Together this reveals that young Australian adults are exposed to health and nutrition information, and product endorsement, primarily from individuals without formal qualifications. Furthermore, the finding in this study that influencers were utilising expert advice as a communication style is in line with previous research finding that the use of language choices to position oneself as an authority figure may enhance engagement [30]. The finding that informal language was the most frequent linguistic feature in analysed posts has similarly been shown in prior research to be a communication strategy that enhances viewer engagement. The authors investigating communication strategies used by OB/GYNs on Chinese social media platforms also found that the use of ‘everyday’ language was one of the three most influential predictors of follower engagement [32]. Such results highlight the complexity of navigating health information on online platforms, with creators employing communication techniques to enhance their appearance of credibility and engagement. The results of this study also suggest there is the potential for health professionals to adopt similar communication styles and linguistic strategies to improve the engagement and reach of high-quality evidence-based nutrition information.

Thematic analysis revealed the overarching concept that users are required to form their own understanding of food and nutrition information on their health journey, within an online environment that includes conflicting ideologies and messaging and is often highly stylised and emotionally charged. This finding adds to the already large body of literature highlighting the challenges of navigating the vast amount of often inaccurate health-related information on social media [18,33]. Additionally, nutrition and health content is constructed using linguistic features that are stylised, often written to evoke emotion from the viewer [12], and found within appealing appearance-focused formats [34]. Given these findings, further attention to and examination of the digital health literacy of young Australian adults would be useful and may help to empower users to engage better with evidence-based, practical health content.

This study has several key strengths and contributes important information on the influence of social media on eating behaviours among young Australian adults. The first strength is the mixed method design, which combines quantitative survey data with content and thematic analysis, thereby strengthening our understanding of the nature of the exposure. This contemporary investigation also fills a critical gap in the evidence base by undertaking a detailed analysis of more than 1400 posts from 71 distinct health content creators identified by study participants. Rather than reviewing generalised exposure, the study thereby sustains ecological validity by analysing real creators and health-related information that participants have access to and engage with. Another key strength is the extensive systematic categorisation of Instagram content into types of content, creator qualification, communication style, and linguistic features. Employing various methodologies for content analysis provided a comprehensive overview of what young Australian adults are consuming in reference to health and nutrition online media.

The mixed-methods design is a key strength of this study. Despite this, there is a limitation regarding how the findings of the two separate study components can be connected. The first part of this study, finding no association between time spent on Instagram and TikTok and eating behaviours, may suggest that the content analysed by the health creators in this study did not influence participants’ eating behaviours. However, due to the observational and cross-sectional nature of this study, this cannot be definitively concluded. Analyses focused on the exposure of total time spent on social media rather than exploring any direct effects of content on long-term eating behaviours. Given that previous research suggests exposure to different types of health content has differential effects on eating behaviours [9], future studies which can more directly assess the effect of exposure to certain content on eating behaviours will be beneficial.

This study has several other weaknesses. First, recruitment strategies resulted in a somewhat homogenous sample, composed predominantly of university-educated participants residing in socioeconomically advantaged areas. As a result, this sample may not accurately reflect the broader diverse population, which has varying degrees of educational attainment and socioeconomic status, introducing sampling bias. Therefore, the findings of this study are limited in their generalisability and more relevant for populations of young adults with higher educational attainment and socioeconomic status and therefore potentially higher digital resilience. Second, the small sample size included in this study results in limited statistical power. This may have contributed to the lack of correlation seen between time spent on social media and SESMEB scores. Therefore, the null findings of this study should be interpreted with caution. Third, a self-administered questionnaire through an online platform introduces potential for both recall bias and social desirability bias. Similarly, although requesting participants to check their phone screen time logs to report time spent on social media for an objective measure, participants may still have self-reported this main exposure, introducing measurement bias. Finally, the cross-sectional nature of this study limits any causal inference about the directional relationship between time on social media or content exposed to and eating behaviours. Further studies with larger, more representative sample sizes and using objective measures of time spent on social media will be beneficial to expand upon the findings of this study.

The key findings of this study have important practical applications for health professionals, particularly dietitians, in clinical, community, and educational settings, as well as for future research. The high frequency of health and nutrition product advertisements on social media highlights the prominence of industry-based marketing and its ability to reach young Australian adults, under the guise of credible health and nutrition content. Furthermore, most content creators lacked formal health qualifications, indicating that young Australians are receiving online health and nutrition information from laypersons without suitable qualifications. Health professionals therefore need to be aware of the types of content that young adult clients are likely exposed to on social media and assess the client’s health literacy and level of engagement with online information. Furthermore, this study highlights the importance of embedding digital health skills in practice to empower clients with strategies to critically evaluate online health information. Dietitians and other health professionals may wish to consider adopting similar linguistic and communication styles when creating their own posts to improve the reach and dissemination of accurate and high-quality nutrition information. In conjunction with previous work by Taba et al. (2025) [35], whereby a framework has been developed for communicating health messages to young people, we suggest more explicit attention from health professionals to this aspect. Moreover, the potential to adopt various communication strategies could be incorporated into dietetic curricula, teaching future dietitians how to engage with digital audiences effectively.

## 5. Conclusions

This study found no association between social media use and eating behaviours amongst a sample of young Australian adults. The qualitative analysis suggests that young adults are frequently exposed to health and fitness product promotions from unqualified content creators on social media. Additionally, health and fitness content creators commonly employed persuasive language strategies to communicate information. Dietitians and nutrition professionals may need to consider adopting specific linguistic and communication styles to enhance dissemination and increase engagement with credible nutrition information online. These findings have implications for improving digital health literacy and strengthening the impact of evidence-based nutrition messaging in digital environments.

## Figures and Tables

**Figure 1 nutrients-18-00044-f001:**
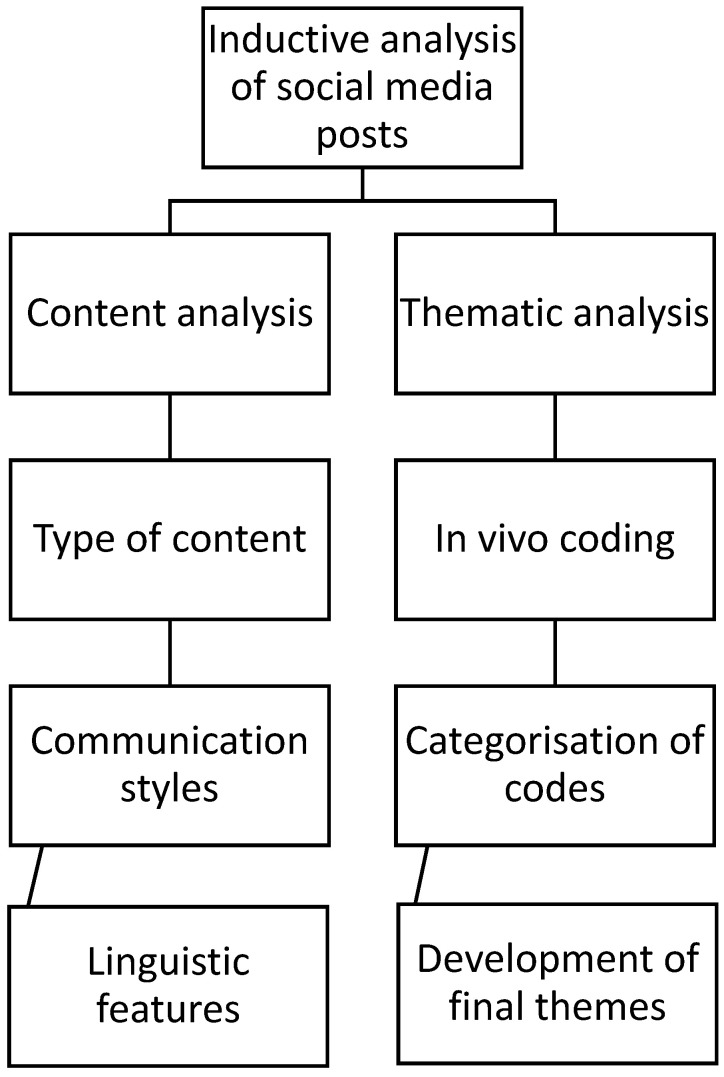
Outline of content and thematic analysis steps undertaken on Instagram posts.

**Figure 2 nutrients-18-00044-f002:**
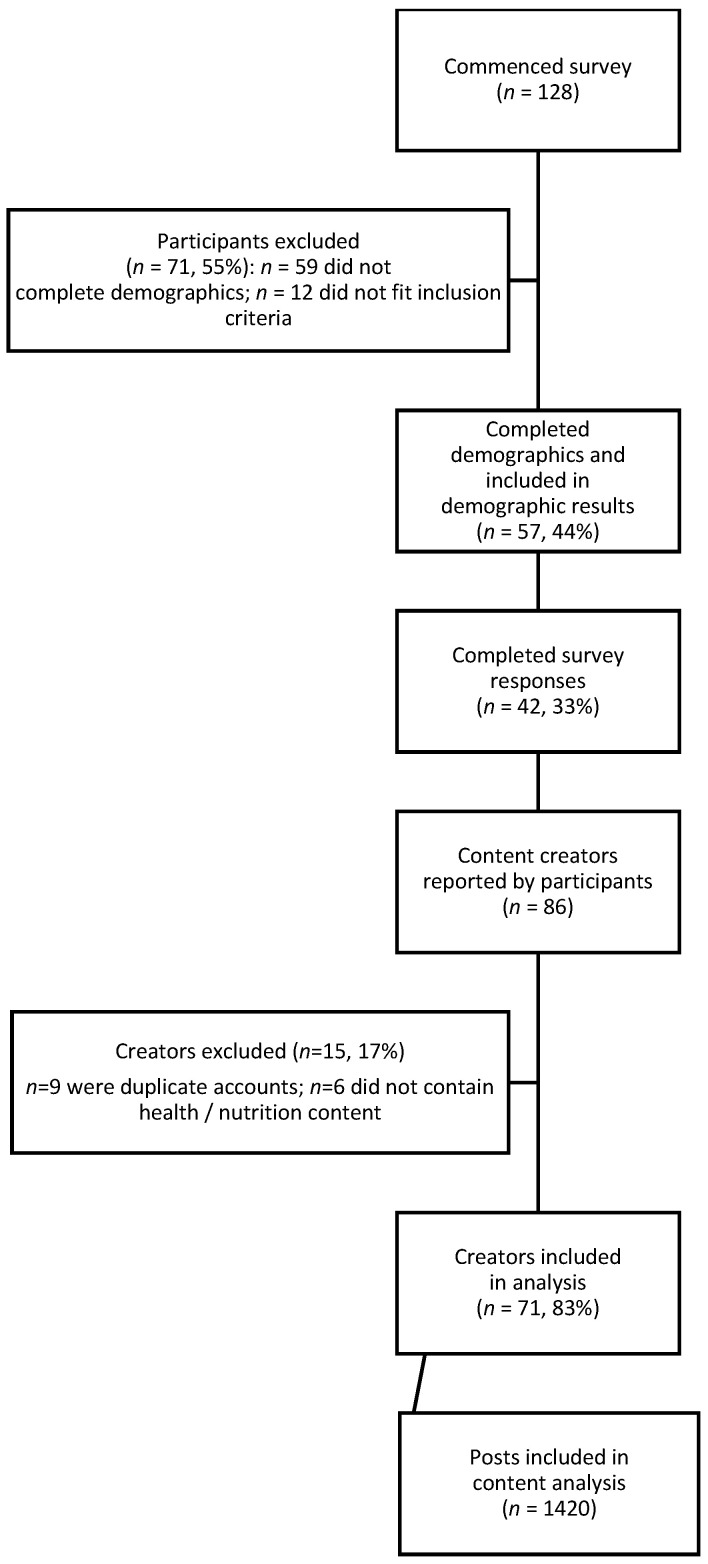
Flowchart of participants and data included in study.

**Figure 3 nutrients-18-00044-f003:**
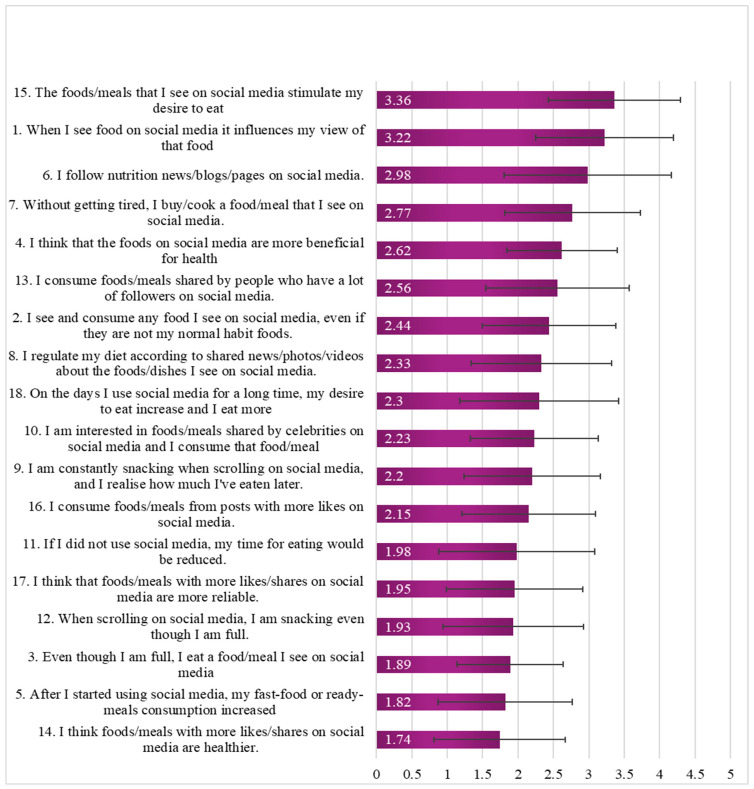
Mean responses to the SESMEB survey ordered from highest to lowest mean score. Numbers correspond to question numbers in the SESMEB questionnaire.

**Table 1 nutrients-18-00044-t001:** Demographic characteristics.

Characteristics	Frequency (*n* = 57)	Percentage (%)
Sex		
Female	38	67.9
Male	17	30.4
Do not wish to disclose	1	1.8
SEIFA rating ^1^		
10	12	21.8
9	4	7.3
8	11	20.0
7	2	3.6
6	16	29.1
4	3	5.5
3	3	5.5
2	2	3.6
1	2	3.6
Highest level of education		
Advanced diploma/diploma	1	1.8
Bachelor’s degree	19	33.3
Trade or certificate (I–IV)	5	8.8
Year 12 or equivalent qualification	32	56.1
Category of post school qualification (*n* = 50)		
Agriculture, environmental, and related studies	1	1.9
Creative arts	3	5.7
Education	4	7.5
Engineering and related technologies	3	5.7
Food, hospitality, and personal services	3	5.7
Health	26	49.1
Management and commerce	2	3.8
Natural and physical sciences	2	3.8
Society and culture	6	11.3
Current main occupation		
Carer and support work	4	8.5
Civil engineer	1	2.1
Cyber security	1	2.1
Education and tutoring	4	8.5
Fitness coach and trainer	3	6.4
Horticulturalist	1	2.1
Hospitality	17	36.2
Lifeguard	1	2.1
Marketing	2	4.2
Nutritionist	1	2.1
Pharmacy	1	2.1
Retail and sales	11	23.1
Special diet followed		
Yes 21 (36.8%)		
Following evidence-based diet (*n* = 11)		
Gluten free	2	4.8
Low FODMAP ^2^	2	4.8
Mediterranean	3	7.1
Vegan	2	4.8
Vegetarian/pescetarian	2	4.8
Following other type of special diets (*n* = 10)		
Carnivore	1	2.4
High protein	1	2.4
Intermittent fasting/time-restricted eating	1	2.4
Low carb	1	2.4
Calorie deficit	4	9.5
Wholefood	2	2.8

^1^ Socioeconomic Index for Areas advantage or disadvantage rating, where 10 indicates the highest advantage. ^2^ Fermentable oligosaccharide, disaccharides, monosaccharides, and polyols.

**Table 2 nutrients-18-00044-t002:** Frequency of content topics posted by Instagram creators.

Type of Content	Frequency (*n* = 71)	Percentage (%)
Ancestral diet nutrition education and dietary advice	2	2.8
Athlete/performance-focused nutrition coaching	2	2.8
Bodybuilding content	2	2.8
Client body transformations	2	2.8
Critiques modern medical practices	3	4.2
Critiques public health paradigms	3	4.2
Discredits online health myths and trends	2	2.8
Display/update of personal physique updates	9	12.7
Event-specific training and preparation	3	4.2
Evidence-based health education advice	4	5.6
Exercise routines and regimes	11	15.5
Fat loss and weight management strategies and advice	10	14.1
Fitness motivation	21	29.6
Fitness/training education, tips, and coaching	10	14.1
Fitness/weight loss journey reflections	8	11.3
General wellness advice and strategies	10	14.1
Health and wellness motivation content	2	2.8
Health/fitness product endorsement/advertisement	40	56.3
Healthy recipes and meal inspiration	30	42.6
Herbal medicine education and advice	2	2.8
High-protein/low-calorie recipes	7	9.9
Holistic health and lifestyle coaching content	5	7.1
Innovative recipe demonstrations	6	8.5
Meal prep recipes	3	4.2
Mental health awareness, personal reflection, or support	4	5.6
Nutrition education and dietary advice	33	46.5
Personal athletic/fitness achievements	7	9.9
Personal fitness goals	2	2.8
Personal health struggles	2	2.8
Plant-based or vegan recipes	5	7.1
Promotion of personal health/fitness products	27	38
Sports and performance nutrition advice	7	9.9
What I eat in a day	5	7.1

**Table 3 nutrients-18-00044-t003:** Details on types of content creators, communication styles, and linguistic features used.

Content Creator Type and Linguistic Features	Frequency	Percentage (%)
Type of Creator (*n* = 71)		
Certified medical professional	2	2.8
Dietitian	8	11.3
Fitness professional	9	12.7
Government organisation	1	1.4
Lay person	40	55.3
Medical doctor	1	1.4
Nutritionist	11	15.5
Dominant Communication Style		
Coaching and mentoring	12	16.9
Expert advice	34	47.9
Intimate experience	26	36.6
Middle-of-the-road	4	5.6
Motivation and guidance	32	45.1
Storytelling	4	5.6
Struggle and overcoming	8	11.3
Linguistic Styles		
Authenticity claims	20	28.2
Entrepreneurship	40	56.3
Humour	32	45.1
Individualism	33	46.5
Informal language	61	85.9
Jargon/slang	58	81.7
Masculinity	29	40.8
Non-conformity	24	33.8
Profanity	15	21.1
Rhetorical question	60	84.5
Shock value	32	45.1
Storytelling	27	38

**Table 4 nutrients-18-00044-t004:** Four key themes identified and example codes from analysed posts.

Key Theme	Example Codes from Posts ^1^
Emotional and psychological content was used to evoke feelings, memories, and psychological responses	“Train your body, but don’t forget that your heart is your strongest muscle. Lift heavy but be kind.” “POV: You show up for yourself every day because that’s what the person you aspire to be would do.” “Never say you can’t do anything. You are stronger than you believe! Find your motivation and don’t look back!” “Guilt doesn’t deserve to be part of Christmas this year. Here’s how you can actually enjoy food this Christmas.” “Sign up for your free trial to workout with me. Let’s feel more connected, challenged, and stronger together!” “We’re so incredibly proud of what we have built and of who we are. Together, women are better.”
Influencers presented themselves as knowledgeable and authoritative	“Race day planning should be tried and tested weeks or even months before your event. Grab a copy of one of our training and nutrition plans to make sure this is your best run yet.” “If you consistently under-consume calories, the body can downregulate processes like digestion to conserve energy.” “This breakfast is packed with protein, healthy fats and fibre and is a great way to stabilise your blood sugar for energy all day.” “It’s so important for our gut to get a wide range of vitamins and minerals.” “If you like chewing gum, skip the ones with aspartame, artificial flavours, and blue1 colouring. Try these better options instead.” “Obviously I’m not female, but I’ve done a bit of research.” “Micronutrients are essential for our bodies and play a HUGE role in brain health, energy metabolism, sleep, mood, and immunity. Micros could be the missing piece if you are feeling sluggish or struggling with recovery!”
Social and cultural framing of content	“Carbs don’t make you fat. Whoever told you that is lying.” “Give yourself permission to include all foods, because food is more than just fuel.” “It’s time to stop labelling foods as good or bad.” “You don’t have to make everything from scratch. I don’t!” “How could these meals be healthy? I’ve recreated each one of your favourite cheat day meals to be lower in calories and higher in protein so you can eat them and still achieve your weight loss goals.” “Aim for a source of protein at every meal and with your snacks to help you stay full, build muscle, and lose fat.”
Social media content was designed to address the practicalities and logistics of managing one’s health and nutrition in everyday life	“Pre-prepared meals can be a super convenient option for when you haven’t had the chance to do meal-prep. Tips: Choose the high protein brands, add frozen veggies to them for extra nutrients and fibre.” “Healthy eating is not black and white: it’s about balance and flexibility. Some days you won’t have the energy to cook a fresh, well-balanced meal and that’s ok! Having options of foods you can throw together for a quick simple meal is important as its about consistency, not perfection.” “It’s all about awareness. Maintaining a healthy weight can be achieved by making sure you don’t consume more calories than you burn day to day.” “We can’t be perfect all the time, but reduce the likelihood of making the wrong choice by planning ahead.”

^1^ All quotes have been reported in a manner to maintain the anonymity of the content creators.

## Data Availability

The original contributions presented in this study are included in the article. Further inquiries can be directed to the corresponding author.

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
