# Peer review of "Unqualified Advice and Product Promotions: Analysis of Health and Nutrition Content on Social Media Consumed by Young Adults"

_nutrients, 2025, doi:10.3390/nu18010044_

Round 1
Reviewer 1 Report
Comments and Suggestions for Authors
The authors discuss a highly topical issue with a clear and up-to-date perspective, in the Australian context.
The structure is clear, the method is well described, and the qualitative analysis is a strong point of the work.
The manuscript is original and well constructed, but requires methodological clarification, more critical discussion, and better contextualisation of its limitations.
The content analysis of 1,420 posts and the classification by communication style and linguistic style are detailed and provide a very comprehensive understanding of the social media landscape.
As already argued by the authors, the small sample size may have attenuated the relationship between social media use and eating behaviours as an effect of digital resilience.
The measurement of social media use is based exclusively on daily time, which is a very rough indicator.
As recent studies show, the type of content matters more than the duration.
The manuscript highlights this in the Discussion, but it is not addressed in the methodology section, further consideration is recommended.
We suggest mitigating the phrase “This study highlights that young adults are primarily exposed...”, which appears too strong for a small sample, and revising the text for greater flow, as it is more descriptive than interpretative.
Author Response
Comment 1: As already argued by the authors, the small sample size may have attenuated the relationship between social media use and eating behaviours as an effect of digital resilience.
The measurement of social media use is based exclusively on daily time, which is a very rough indicator.
Response 1: Thank you for highlighting the need to discuss these limitations in further detail. We have added more discussion about the limitations of sample size and sampling bias in the discussion. We have also mentioned the limitation of self-reported time spent on social media as the main exposure, and recommended future studies have larger sample size and use more objective measures. See Lines 494-511.
Comment 2: As recent studies show, the type of content matters more than the duration. The manuscript highlights this in the Discussion, but it is not addressed in the methodology section, further consideration is recommended.
Response 2: The first aim of this study was to examine the correlation between duration of social media use and eating behaviours. The second aim was to examine the health and nutrition content from social media posts. We didn’t explore the relationship between exposure to different types of content and influence on eating behaviours. This is therefore a point raised in the discussion and further research ought to examine whether exposure to different content has an influence on long-term eating behaviours.
Comment 3: We suggest mitigating the phrase “This study highlights that young adults are primarily exposed...”, which appears too strong for a small sample, and revising the text for greater flow, as it is more descriptive than interpretative.
Response 3: Word “highlight” has been replaced with “suggest” at Line 31.
The discussion section has been reviewed to ensure the findings of this study are interpreted and discussed in the context of the broader literature. As a result, we have made changes to third paragraph to enhance the discussion (Lines 441-454).
Reviewer 2 Report
Comments and Suggestions for Authors
The article titled “nutrients-4024594_Unqualified Advice and Product Promotions: Does Nutrition Messaging on Social Media Influence Eating Behaviours in Young Adults?” is presented in the “Nutrition and Public Health” section of the magazine “Nutrients”.
This study explored how time spent on Instagram and TikTok relates to eating behaviors among young Australian adults, and examined the types of content and communication styles used by health and nutrition creators on Instagram. Among the 42 participants who completed all study measures, no significant association was identified between time spent on these platforms and SESMEB eating-behavior scores. The analysis of 1,420 Instagram posts showed that most content consisted of product promotions from non-expert creators, often framed as “expert advice” and conveyed using informal language. The thematic analysis indicated that online health information is frequently emotionally charged, stylized, and at times contradictory, requiring users to navigate conflicting messages. Overall, the findings suggest that dietitians and nutrition professionals may need to adopt more engaging linguistic and communication strategies to enhance digital health literacy and strengthen the impact of evidence-based nutrition information online.
Comments
This manuscript addresses a highly relevant and timely topic—the relationship between social media use and eating behaviors, alongside an in-depth analysis of the nature of health and nutrition content shared on Instagram. The mixed-methods design is appropriate for the study's objectives, and the qualitative component is particularly robust. The manuscript is clearly written and provides valuable insights for nutrition professionals and for digital health communication. However, several methodological and interpretive issues require attention to strengthen the validity and clarity of the findings.
Major Weaknesses and Required Revisions
- Small and non-representative sample
Only 42 participants completed the full study, which substantially limits statistical power. The manuscript does not indicate whether a sample-size calculation was carried out. Recommendation: Explicitly acknowledge the limitations of the small sample size in relation to the null findings and discuss potential sampling bias, given that all participants were individuals already seeking health or nutrition content online.
- Limited inference due to cross-sectional design
The manuscript occasionally implies directional or causal relationships between exposure and eating behaviors. This design can only describe associations. Recommendation: Review any statements that suggest causality and emphasize the observational, cross-sectional nature of the study.
- Self-reported exposure measures
Time spent on Instagram and TikTok was self-reported, raising concerns about accuracy. It is unclear whether these measures were validated. Recommendation: Expand the discussion of measurement bias and consider recommending that future research incorporates objective metrics (e.g., device-generated screen-time logs).
- Narrow platform focus
Excluding other widely used platforms such as YouTube or Facebook may limit generalisability. Recommendation: Provide a rationale for focusing solely on Instagram and TikTok, or discuss how differences between platforms may have influenced the findings.
- Characterization of content creators The classification of creators as “laypersons” based primarily on profile cues may oversimplify issues of expertise and credibility.
Recommendation: Offer clearer criteria for defining creator qualifications or recognize the limitations of this categorization.
- Integration between quantitative and qualitative components
Although both methodological components are well presented, the manuscript would benefit from clearer integration of quantitative and qualitative insights.
Recommendation: Add a dedicated section in the Discussion explicitly connecting and contrasting the two strands.
Minor Comments
Clarify whether SESMEB subscale scores were analyzed or only the total score.
Provide more detail on how inter-coder reliability was assessed during qualitative coding.
Ensure consistency in terminology when describing communication styles.
Improve the structure of the Results section by clearly separating descriptive findings from interpretative commentary.
Author Response
Major Weaknesses and Required Revisions
- Small and non-representative sample
Only 42 participants completed the full study, which substantially limits statistical power. The manuscript does not indicate whether a sample-size calculation was carried out. Recommendation: Explicitly acknowledge the limitations of the small sample size in relation to the null findings and discuss potential sampling bias, given that all participants were individuals already seeking health or nutrition content online.
Response 1: Thank you for this recommendation. We have now explicitly mentioned the small sample size as a limitation on lines 501-503. We have also expanded the mention of sampling bias related to the education and socioeconomic status seen in this study sample (Lines 494-501).
2. Limited inference due to cross-sectional design
The manuscript occasionally implies directional or causal relationships between exposure and eating behaviours. This design can only describe associations. Recommendation: Review any statements that suggest causality and emphasize the observational, cross-sectional nature of the study.
Response 2: Thank you for pointing this out. We have replaced the word “relationship” with “association” on Lines 412 and 535 and have included the cross-sectional nature of this study as a limitation at lines 508-510. We have additionally updated the title of this study to better reflect the aims of the project and minimise assumptions of causality between content exposure and eating behaviours.
3. Self-reported exposure measures
Time spent on Instagram and TikTok was self-reported, raising concerns about accuracy. It is unclear whether these measures were validated. Recommendation: Expand the discussion of measurement bias and consider recommending that future research incorporates objective metrics (e.g., device-generated screen-time logs).
Response 3: We did request that participants check their phone screen-time logs to obtain an accurate report of time spent on Instagram and Tiktok (Lines 112-114). However, we cannot guarantee the accuracy of participants reporting . Therefore, we have discussed the limitation of self-reporting the main outcome in the discussion on lines 505-507.
4. Narrow platform focus
Excluding other widely used platforms such as YouTube or Facebook may limit generalisability. Recommendation: Provide a rationale for focusing solely on Instagram and TikTok, or discuss how differences between platforms may have influenced the findings.
Response 4: Thank you for this suggestion. We mentioned that engagement with nutrition and health content is popularly promoted in short-form videos on Tik Tok and Instagram and therefore these platforms were the focus. We have now justified this choice of assessing these platforms in the methods section (Lines 116-117).
5. Characterization of content creators. The classification of creators as “laypersons” based primarily on profile cues may oversimplify issues of expertise and credibility.
Recommendation: Offer clearer criteria for defining creator qualifications or recognize the limitations of this categorization.
Response 5: Thank you for highlighting this. We have added brief further clarification as to how creators were considered on Lines 158-161 of the methods as below:
“These categories were developed after reviewing the creators nominated by participants and identifying whether they had relevant evidence-based health, nutrition or dietetic educational qualifications. Content creators were categorized as laypersons when they had not listed any educational qualifications, for example University degree, in their Instagram biography..”
6. Integration between quantitative and qualitative components
Although both methodological components are well presented, the manuscript would benefit from clearer integration of quantitative and qualitative insights. Recommendation: Add a dedicated section in the Discussion explicitly connecting and contrasting the two strands.
Response 6: We have added a paragraph in the Discussion connecting the two separate components of the study and addressing limitations of drawing direct conclusions from these two study components. See Lines 482-492
Minor Comments
- Clarify whether SESMEB subscale scores were analyzed or only the total score
Response: We have now clarified this on line 129 – only total scores were analysed.
- Provide more detail on how inter-coder reliability was assessed during qualitative coding.
Response: Content analysis was undertaken by one researcher only and therefore inter-coder reliability not relevant. This has been specified more clearly on line 149.
Inductive thematic analysis was undertaken by two researchers. The process of collaboratively deciding on themes has been specified on lines 185-187. Additionally, to aid in understanding of the methods, we have added in a figure (Figure 1) to briefly outline the steps of content and thematic analysis undertaken in this study.
- Ensure consistency in terminology when describing communication styles.
Response: Thank you for this suggestion. We have reviewed the manuscript and have not identified any discrepancies when referring to the types of communication styles or linguistic features.
- Improve the structure of the Results section by clearly separating descriptive findings from interpretative commentary.
Response: We have reviewed the results section and made a minor change to wording of the third theme identified on lines 373-373 however have not identified any other sections in the results containing interpretative commentary. Section 3.4 is a description of the themes and concepts identified from thematic analysis.
Round 2
Reviewer 2 Report
Comments and Suggestions for Authors
Comments:
After reviewing the authors' responses and the revised manuscript, I conclude that the majority of the requested revisions have been addressed appropriately. The authors have strengthened the methodological transparency, corrected causal language, improved justification for the platform selection, clarified the criteria for classifying content creators, and enhanced the integration between the quantitative and qualitative components. However, a few points still require further refinement:
- The small sample size remains a major limitation of the study. Although it is now mentioned in the manuscript, this limitation should be more clearly linked to the restricted statistical power and the caution needed when interpreting null findings.
- The Discussion should explicitly recommend the use of objective screen-time measures in future research.
- The Results section may still benefit from a clearer distinction between descriptive reporting and interpretative commentary.
Author Response
After reviewing the authors' responses and the revised manuscript, I conclude that the majority of the requested revisions have been addressed appropriately. The authors have strengthened the methodological transparency, corrected causal language, improved justification for the platform selection, clarified the criteria for classifying content creators, and enhanced the integration between the quantitative and qualitative components. However, a few points still require further refinement:
- The small sample size remains a major limitation of the study. Although it is now mentioned in the manuscript, this limitation should be more clearly linked to the restricted statistical power and the caution needed when interpreting null findings.
Response:We had already stated this in the limitation:
“Second, the small sample size included in this study results in limited statistical power. This may have contributed to the lack of correlation seen between time spent on social media and SESMEB scores”
For further clarification we have added the following sentence on Line 517/518:
“Therefore, the null findings of this study should be interpreted with caution.” - The Discussion should explicitly recommend the use of objective screen-time measures in future research.
Response:This has already been stated on Lines 525-527:
“Further studies with larger, more representative sample sizes and using objective measures of time spent on social media will be beneficial to expand upon the findings of this study.” - The Results section may still benefit from a clearer distinction between descriptive reporting and interpretative commentary.
Response:There is a component of the 6-step thematic analysis framework we used that involves interpreting the key themes from the identified codes. The themes are actively constructed by researchers through a reflexive and interpretative process. The results in section 3.4 are interpretive in this regard. No other results presented are interpretative. However section 3.4 thematic analysis results are reported in line with how thematic analysis results are commonly presented.
See the following similar papers for example:Sarapis, K.; Cao, Y.; Abou Chakra, M.; Nunn, J.; Rathod, P.; Weber, M.; Albuquerque, C.; Chapman, M.; Barr, R.; Gilfillan, C.; et al. Understanding the Unmet Needs of People Living with Type 2 Diabetes in Self-Managing Their Condition. Nutrients 2025, 17, 1243. https://doi.org/10.3390/nu17071243
McLean, C.P.; de Boer, K.; Lee, M.F.; McLean, S.A. The Treatment Experiences of Vegetarians and Vegans with an Eating Disorder: A Qualitative Study. Nutrients 2025, 17, 345. https://doi.org/10.3390/nu17020345
Therefore, we are unsure of a different way to present the results of the thematic analysis in response to your recommendation.